# Genetic and Methylation Analysis of CTNNB1 in Benign and Malignant Melanocytic Lesions

**DOI:** 10.3390/cancers14174066

**Published:** 2022-08-23

**Authors:** Anne Zaremba, Philipp Jansen, Rajmohan Murali, Anand Mayakonda, Anna Riedel, Dieter Krahl, Hans Burkhardt, Stefan John, Cyrill Géraud, Manuel Philip, Julia Kretz, Inga Möller, Nadine Stadtler, Antje Sucker, Annette Paschen, Selma Ugurel, Lisa Zimmer, Elisabeth Livingstone, Susanne Horn, Christoph Plass, Dirk Schadendorf, Eva Hadaschik, Pavlo Lutsik, Klaus Griewank

**Affiliations:** 1Department of Dermatology, University Hospital Essen, University of Duisburg-Essen, Hufelandstr. 55, 45122 Essen, Germany; 2German Cancer Consortium (DKTK), 69120 Heidelberg, Germany; 3Department of Pathology, Memorial Sloan Kettering Cancer Center, New York, NY 10065, USA; 4Division of Cancer Epigenomics, German Cancer Research Center (DKFZ), 69120 Heidelberg, Germany; 5Helmholtz International Graduate School for Cancer Research, 69120 Heidelberg, Germany; 6Faculty of Biosciences, Heidelberg University, 69120 Heidelberg, Germany; 7Institute for Dermatohistopathlogy, 69120 Heidelberg, Germany; 8Dermatology, 66482 Zweibrücken, Germany; 9Dermatology, 66346 Püttlingen, Germany; 10Department of Dermatology, Venereology and Allergology, University Medical Center and Medical Faculty Mannheim, Heidelberg University, 69117 Mannheim, Germany; 11Rudolf-Schönheimer-Institute of Biochemistry, Medical Faculty, University Leipzig, 04109 Leipzig, Germany

**Keywords:** deep penetrating nevus, deep penetrating melanoma, malignant melanoma, mutation profiling, immune checkpoint inhibition

## Abstract

**Simple Summary:**

Recurrent *CTNNB1* exon 3 mutations have been recognized in the distinct group of melanocytic tumors showing deep penetrating nevus-like morphology and in 1–2% of advanced melanoma. We performed a detailed genetic analysis of difficult-to-classify nevi and melanomas with *CTNNB1* mutations and found that benign tumors (nevi) show characteristic morphological, genetic and epigenetic traits, which distinguish them from other nevi and melanoma. Malignant *CTNNB1*-mutant tumors (melanoma) demonstrated a different genetic profile, grouping clearly with other non-*CTNNB1* melanomas in methylation assays. To further evaluate the role of *CTNNB1* mutations in melanoma, we assessed a large cohort of clinically sequenced melanomas, identifying 38 tumors with *CTNNB1* exon 3 mutations, including recurrent S45 (*n* = 13, 34%), G34 (*n* = 5, 13%), and S27 (*n* = 5, 13%) mutations. Locations and histological subtype of *CTNNB1*-mutated melanoma varied; none were reported as showing deep penetrating nevus-like morphology. The most frequent concurrent activating mutations were *BRAF* V600 (55%) and *NRAS* Q61 (34%).

**Abstract:**

Melanocytic neoplasms have been genetically characterized in detail during the last decade. Recurrent *CTNNB1* exon 3 mutations have been recognized in the distinct group of melanocytic tumors showing deep penetrating nevus-like morphology. In addition, they have been identified in 1–2% of advanced melanoma. Performing a detailed genetic analysis of difficult-to-classify nevi and melanomas with *CTNNB1* mutations, we found that benign tumors (nevi) show characteristic morphological, genetic and epigenetic traits, which distinguish them from other nevi and melanoma. Malignant *CTNNB1*-mutant tumors (melanomas) demonstrated a different genetic profile, instead grouping clearly with other non-*CTNNB1* melanomas in methylation assays. To further evaluate the role of *CTNNB1* mutations in melanoma, we assessed a large cohort of clinically sequenced melanomas, identifying 38 tumors with *CTNNB1* exon 3 mutations, including recurrent S45 (*n* = 13, 34%), G34 (*n* = 5, 13%), and S27 (*n* = 5, 13%) mutations. Locations and histological subtype of *CTNNB1*-mutated melanoma varied; none were reported as showing deep penetrating nevus-like morphology. The most frequent concurrent activating mutations were *BRAF* V600 (*n* = 21, 55%) and *NRAS* Q61 (*n* = 13, 34%). In our cohort, four of seven (58%) and one of nine (11%) patients treated with targeted therapy (BRAF and MEK Inhibitors) or immune-checkpoint therapy, respectively, showed disease control (partial response or stable disease). In summary, *CTNNB1* mutations are associated with a unique melanocytic tumor type in benign tumors (nevi), which can be applied in a diagnostic setting. In advanced disease, no clear characteristics distinguishing *CTNNB1*-mutant from other melanomas were observed; however, studies of larger, optimally prospective, cohorts are warranted.

## 1. Introduction

Studies over the past decade have yielded a more detailed understanding of the genetics of melanocytic tumors. Activating mutations of the MAPK pathway are present in most nevi and melanomas. Genetically, melanoma has been classified based on driver mutations activating the MAPK signaling pathway as (I) *BRAF*-mutant (50–60%), (II) *RAS-*mutant (20–30%), (III) *NF1*-mutant (10–15%) or (IV) triple (*BRAF*, *NRAS* and *NF1*) wild-type melanoma (~10%) [1,2]. Cutaneous melanoma harbors a larger number of mutations than any other major cancer entity as a result of UV-exposure. In both, benign and malignant melanocytic tumors, mutations in conjunction with MAPK mutations can occur. Another pathway playing a critical role in melanocyte biology is the Wnt/β-catenin signaling pathway. Inactivation of β-catenin (*CTNNB1*) in neural crest cells during embryogenesis can prevent development of the melanocytic lineage [3]. Activation of the pathway promotes both differentiation and expansion from neural crest progenitors to melanocytes [4]. Activating mutations of *CTNNB1* have been reported in a range of cancers including melanoma [5]. Intracellular levels of CTNNB1 control canonical Wnt/β-catenin signaling. In a physiological setting, in the presence of Wnt ligands, CTNNB1 is translocated to the nucleus and activates the transcription of downstream target genes by binding with members of the Lef/Tcf transcription factors family [6,7]. In the absence of Wnt ligands, cytoplasmic CTNNB1 is targeted by a destruction complex that phosphorylates highly conserved serine/threonine residues located in exon 3 of *CTNNB1*, leading to degradation by the proteasome [7].

In melanocytic tumors, *CTNNB1* mutations have been found to be present in almost all cases of deep penetrating nevi [8]. A recent review summarizing data from multiple next generation sequencing (NGS) approaches found only eight *CTNNB1* mutations in 686 melanomas (1.2%). In this study, *CTNNB1* mutations only occurred in *BRAF* or *NRAS* mutated melanomas, suggesting a cooperation between MAPK and Wnt/β-catenin signaling pathways [9]. Another large retrospective study including NGS data from 467 melanoma patients identified ten primary melanoma patients harboring a *CTNNB1* mutation [7]. Here, concurrent *CTNNB1* and *MAPK* mutations were found to not necessarily confer a deep penetrating nevi phenotype, and often progress to a metastatic stage [7]. The role of Wnt/β-catenin signaling in melanoma remains controversial. Although CTNNB1 has been reported to induce melanoma metastasis [10], it has also been described to limit invasion of melanoma in an experimental setting in both human and mice [11].

Deep penetrating nevi are rare, and even though most can be easily recognized by a well-trained pathologist, they can, on occasion, be mistaken for melanoma. In cases of deep penetrating melanocytic proliferations in which definitive diagnosis by morphology and immunohistochemistry (IHC) alone is difficult, molecular assays including screening for presence of *CTNNB1* mutations as well as other alterations such as *CDKN2A* loss or *TERT* promoter mutations [12,13], may provide additional information to aid classification. Accurately classifying melanocytic tumors as benign or malignant has important implications for the patient in terms of prognosis, follow-up and treatment (including newly introduced adjuvant therapies, as targeted BRAF-inhibitors and MEK-inhibitors [14,15,16,17,18] and immune checkpoint therapies [19,20] which have shown great promise). However, many melanomas still fail to respond to therapy or develop resistance to initially effective therapies [21,22]. A better understanding of which tumors will respond to therapy and how to identify and circumvent resistance in tumors would be of great clinical benefit to affected patients.

In this retrospective study, our aim was to explore to what extent *CTNNB1* mutations are associated with certain clinical, histological, genetic, epigenetic and therapeutic features in melanocytic tumors. Mutation, copy number and methylation profiles were evaluated in *CTNNB1-*mutant melanocytic tumors and compared with those of *CTNNB1*-wild type tumors. In addition, the role of *CTNNB1* mutations in a large cohort of advanced melanoma was analyzed to study potential associations with clinical characteristics, outcome and therapy responses [23].

## 2. Materials and Methods

### 2.1. Patient Selection

Patient medical history and data were retrieved from the medical databases/documentation system of the University Hospital Essen. The study was approved by and performed in accordance with the guidelines of the ethics committee of the University of Duisburg-Essen (BO-9589-20). Molecular testing was performed in patients included, with informed consent. To address all aims, three distinct cohorts were studied. (I) A cohort of seven *CTNNB1*-mutant melanocytic tumors was analyzed using genome-wide DNA methylation in conjunction with copy number variation (CNV) and mutation profiling (Section 3.1 and Section 3.2). The control group consisted of eight benign nevi, eight malignant melanoma, and five Spitz nevi. Clinical characteristics of patients from the control group are shown in Appendix A. Additional information regarding these cases can be found in a previous manuscript [24]. The seven difficult to classify cases were either seen at our department or referred to our department from other institutions for further analysis. Tumors with deep-penetration nevus-like morphology characteristically show large cells with no maturation toward deeper tissue and an infiltrative growth pattern, expanding in an interstitial fashion into the tissue. These are traits that can also be seen in melanoma. In immunohistochemistry, deep penetrating nevus (DPN) like tumors can express HMB45 at certain amounts and demonstrate some level of reactivity to MIB (or Ki-67). All cases deemed not clearly benign by conventional histopathological analysis were passed on for genetic analysis.

(II) Clinical and mutational data from *n* = 38 *CTNNB1*-mutant melanoma patients were retrieved from routinely performed NGS melanoma panel analysis and the medical databases/documentation system of the University Hospital Essen (Section 3.3 and Section 3.4). Only melanoma patients with metastatic disease (advanced melanoma) and therefore clinically confirmed malignant disease in stages IIIA or higher were included (see also Table 1). (III) Anti-PD1-treated melanoma patients were retrieved from [23]. Chi square and Fisher t tests were used for comparison of categorical variables as applicable. A Kruskal–Wallis test was used for continuous variables. Wilcoxon rank sum tests were used for continuous variables in R 4.2.0 (Section 3.5). For survival analysis, progression-free survival (PFS) and overall survival (OS) were defined as time from therapy start, until disease progression or death, respectively; if no such event occurred, the date of the last patient contact was used as endpoint of survival assessment (censored observation).

### 2.2. Histopathology and Immunohistochemistry

Formalin-fixed, paraffin-embedded (FFPE) tissue samples were obtained. For histopathologic examination, 2 µm-thin sections were routinely stained with hematoxylin-and-eosin. Additional immunohistochemical staining was performed, including e.g., MelanA (1:100, Dako, Glostrup, Denmark; M7196) and Ki-67/MIB1 (1:200, Zytomed, Berlin, Germany; MSK0810). All histologic and IHC sections were reviewed by at least two board-certified dermatopathologists (EH, KG).

### 2.3. DNA Isolation and Targeted Sequencing

Sections of 10 µm-thick, were cut from formalin-fixed, paraffin embedded tumor tissues. The sections were deparaffinized and manually microdissected according to standard procedures. Genomic DNA was isolated using the Lotus DNA Library Prep Kit from IDT^®^, according to the manufacturer’s instructions. DNA capture-based, targeted next-generation sequencing was applied to analyze 611 target genes known to be mutated in cutaneous melanoma and other human cancers (see Appendix A for full gene lists) in all samples where sufficient amounts of DNA were available. All samples were sequenced applying a smaller 36 gene panel covering the most relevant melanoma genes (Appendix A). Both sequencing panels were applied using the GeneRead Library Prep Kit from QIAGEN according to the manufacturer’s instructions. Adapter ligation and barcoding was performed, applying the NEBNext Ultra DNA Library Prep Mastermix Set and NEBNext Multiplex Oligos for Illumina from New England Biolabs. IDT for Illumina TruSeq DU Indexes from Illumina^®^ were used and 24 samples run in parallel. The 611 gene panel was sequenced on an Illumina^®^ NextSeq 2000 sequencer the 36 gene panel on an Illumina^®^ MiSeq sequencer.

### 2.4. Mutation Sequence Analysis

CLC Cancer Research Workbench from QIAGEN^®^ was used for sequence analysis, as has been previously reported [25,26,27]. In brief, the analysis workflow described included adapter trimming and read pair merging, before mapping to the human reference genome (hg19). Detection of insertions and deletions as well as single nucleotide variants followed. Additional information regarding potential mutation type, known single nucleotide polymorphisms and conservation scores was obtained by cross-referencing various databases (COSMIC, ClinVar, dbSNP, 1000 Genomes Project, HAPMAP, and PhastCons-Conservation_scores_hg19). Further analysis of csv files was performed by applying R (version 4.0.3). The mean coverage of the targeted sequencing region achieved in targeted DNA sequencing of all *CTNNB1* mutant samples 1776.3 reads with 99.9% of the target region sequenced with a coverage ≥30 reads. Mutations were considered if coverage of the mutation site was ≥30 reads, ≥10 reads reported the mutated variant and the frequency of mutated reads was ≥10%. Copy number variations determined by targeted sequencing were detected with CLC Cancer Research Workbench (QIAGEN^®^) and are based on the following algorithms [28,29,30]. A ≥1.7 absolute fold copy number change involving a region with greater than 30 target sequences was chosen as a cut-off for detecting copy number variations.

### 2.5. DNA-Methylation Profiling and Copy Number Analysis

Array-based copy number and methylation analysis required 500 ng of isolated DNA and was performed on (Spitz nevi (*n* = 5), benign nevi (*n* = 8), malignant melanoma (*n* = 8), and *CTNNB1* mutant melanocytic tumors (*n* = 7). The HumanMethylationEPIC (EPIC) bead-based microarrays from Illumina were used to obtain genome-wide methylation data, according to the manufacturer’s instructions [31,32]. Methylation analysis using EPIC arrays was performed by the Genomics and Proteomics Core Facility, Heidelberg, Germany. Unnormalized DNA methylation data were obtained as IDAT files, which were used as input to the RnBeads software package implementing a comprehensive workflow for quality control, preprocessing and analysis of data from DNA methylation microarrays [32,33]. In brief, DNA-methylation data were normalized by performing background correction and dye bias correction, whereupon low-quality and potentially biased measurements, e.g., from probes obtained with too few microarray beads, probes with low signal/noise ratio (detection p-value), probes containing single nucleotide polymorphisms and cross-reactive probes, are removed or masked. 10,000 sites, most variable across all samples were used for both, principal component analysis and clustering analysis and visualized as heatmaps [32,34]. The copy number profile was generated from the array data using the “conumee” R package in Bioconductor (http://www.bioconductor.org/packages/release/bioc/html/conumee.html (package version 1.26.0), accessed on 20 June 2022). The conumee median signals per bin were summarized in chromosome arms, and the gains and losses were called for arms with summarized median signal above 0.1 or below −0.1, respectively.

### 2.6. Reference-Free Methylome Deconvolution Using MeDeCom

DNA methylation data of the bulk tumor samples of the cohort were investigated using the reference-free MeDeCom algorithm that dissects DNA methylation data into major components of variation, called latent methylation components (LMC) [35]. DNA methylation data of patients were processed according to a recently published protocol [36]. The protocol selected the 20,000 most variably methylated CpG sites across the samples as input to MeDeCom. LMCs were functionally annotated within the FactorViz platform [36].

## 3. Results

### 3.1. CTNNB1 Mutations in Difficult-to-Classify Benign and Malignant Melanocytic Tumors

Challenging melanocytic cases sent for reference histopathology from experienced dermatohistopathologists were sequenced using NGS. Seven cases with *CTNNB1* mutations were identified, two classified as malignant, one as a “deep blue nevus-like melanoma” (case 3) and a “malignant melanoma morphologically resembling a deep penetrating blue nevus” (case 6). The other cases had been classified as a mainly intradermal Spitz nevus (case 1), a deep penetrating blue nevus (case 2), a combined nevus (case 4), a deep-penetrating nevus (case 5), and a congenital nevus cell nevus (case 7) (Table 1). Six patients were male and two tumors were localized on the head. Two tumors harbored additional BRAF V600E mutations and one showed an NRAS Q61R mutation. Both melanomas showed mutations in the TERT promoter region, and one additionally harbored an NF1 mutation (Table 1). (Full clinical details of the two melanoma cases are included in the Appendix A).

### 3.2. Methylation Profiling with Comprehensive Copy Number Analysis

Methylation analysis was performed on the seven described *CTNNB1*-mutant tumors and the profiles were compared to control groups consisting of eight benign nevi, eight malignant melanomas and five Spitz nevi [24] (Figure 1A). HumanMethylationEPIC data were preprocessed using RnBeads workflow to obtain genome-wide DNA methylation profiles [32]. To assess the global trends of DNA methylation variability of *CTNNB1*-mutant melanocytic tumors, we applied principal components analysis (PCA). We observed a distinction between the histopathological entities (nevi, Spitz nevi, and melanoma) within the first two principal components (Figure 1B). *CTNNB1*-mutant melanocytic tumors showed a methylation profile clustering between Spitz and normal nevi. The two malignant *CTNNB1*-mutant tumors (ID 3 + 6) clustered towards the melanoma group. Hierarchical clustering analysis of average methylation profiles of 10,000 gene regions confirmed two distinct groups: (1) benign *CTNNB1*-mutant melanocytic tumors, nevi, Spitz nevi, and (2) malignant *CTNNB1*-mutant tumors and *CTNNB1*-wild type melanoma (Figure 1D). A clear separation was also observed regarding CNV, which were not detected in benign *CTNNB1*-mutant tumors, but were present in *CTNNB1*-mutant melanoma (Figure 1B, Table 1). To better understand cellular composition of *CTNNB1* mutant samples and control groups, bulk DNA methylation data were deconvoluted assessed using reference-free algorithm MeDeCom that dissects DNA methylomes into major components of variation, called latent methylation components (LMC), and estimates their relative proportions. The cross-validation error pointed at five LMCs (parameter k) during MeDeCom parameter selection and the regularization parameter λ value of 0.01 (Appendix A). The proportions of the five LMCs (LMC1-5) in all samples are visualized in Figure 1E. Hierarchical clustering analysis of LMC proportions revealed well-separated clusters corresponding to the benign and malignant tumors (Figure 1E).

### 3.3. Clinical Characteristics of CTNNB1 Mutated Melanoma Patients

Data acquired from routinely performed NGS panel analysis of histopathologically clearly diagnosed melanomas analyzed between 2014 and 2021 were assessed to identify samples harboring *CTNNB1* mutations. Mutations were found to be distributed across the gene, however recurrent mutations were found in the known N-terminal hotspot region on exon 3 ranging from amino acid 25 to 46 (Figure 2A,B) [5,37]. The remainder of the identified mutations were distributed randomly across the gene and are assumed to be passenger mutations (Figure 2A,B). In total, 38 mutations in melanoma from 38 patients in the exon 3 hotspot of CTNNB1 were identified. The majority of patients (*n* = 27, 71%) were male, and the median age at melanoma diagnosis was 59 years (range 39–90 years). Eighteen percent (*n* = 7) of patients had a nodular melanoma, 13% (*n* = 5) each a superficial spreading melanoma or a melanoma of unknown primary (MUP), 8% (*n* = 3) an acrolentiginous melanoma (ALM), 3% (*n* = 1) had a spitzoid melanoma, and 45% (*n* = 17) had an unspecified histopathological subtype (Table 2). Forty-eight percent (*n* = 18) of primary melanoma were ulcerated. Primary melanomas were localized on the lower extremity in 37% (*n* = 14), trunk (32%, *n* = 12) and the head/neck region (13%, *n* = 5) (Table 2). Forty-two percent of patients (*n* = 16) were stage IIIA or higher at diagnosis. During the disease course, 66% (*n* = 25) of patients with a CTNNB1 mutation developed lymph node metastasis, 32% (*n* = 12) lung metastasis, 29% (*n* = 11) hepatic metastasis, 29% (*n* = 11) metastasis within the central nervous system (CNS), and 11% (*n* = 4) bone metastasis. Nine patients (24%) each received either targeted therapy (TT) or immune checkpoint inhibitor-based therapy as a first-line treatment. Patients treated with TT showed partial response (PR, *n* = 2), stable disease (SD, *n* = 2) and progressive disease (PD, *n* = 3). In two cases the response was unknown. Patients treated with immune checkpoint inhibitor-based therapy showed PR (*n* = 1) and PD (*n* = 8). Other therapies administered for advanced disease included chemotherapy (*n* = 1, PD), the NIPAWILMA trial (*n* = 1, PD) [38], and the TriN 2755 trial (*n* = 1, SD) [39].

### 3.4. Mutations within CTNNB1 in Melanoma

All patients showed somatic mutations of *CTNNB1* in the protein hotspot region of exon 3 between W25 and G48 (Figure 2A,B, Table 3). In 12/38 patients, multiple mutations within the *CTNNB1* gene were present. Most frequent mutations were present at S45 (*n* = 13, 34%), G34 (*n* = 5, 13%), and S27 (*n* = 5, 13%). All mutations are shown in Table 3. Fifty-five percent of patients (*n* = 21) had concurrent mutations in *BRAF* V600 (18 V207E, 1 V207D and 2 V207R) and 34% (*n* = 13) in *NRAS* Q61 (5 Q61K, 3 Q61R, 2 Q61L, 2 Q61H, and 1 Q61V) (Figure 2C). The majority (76%, *n* = 29) of patient samples had mutations in the *TERT* promoter region (Table 3).

### 3.5. CTNNB1 Mutation Status and Transcriptomic Alterations in an Anti-PD1 Monotherapy Treated Melanoma Cohort

A previously described cohort of 144 melanoma patients treated with anti-PD1 monotherapy and mutational and transcriptomic data were used to investigate therapy response and gene expression profiles in *CTNNB1* mutant patients (23). Nine (6.2%) showed a mutation in exon 3 of *CTNNB1*, of which transcriptomic data were available in eight cases. No significant differences within clinical characteristics and disease course were found between *CTNNB1* mutant and non-mutant patients (Table 4). Therapy response to anti-PD1 monotherapy showed an overall response rate (ORR) of 67% (6/9 patients) in patients with *CTNNB1*-mutant melanoma compared to 36% (49/135 patients) in those with *CTNNB1*-wild type tumors. Median number of total mutations, nonsynonymous mutations, clonal and subclonal mutations were higher in *CTNNB1* mutant melanoma patients, albeit not significantly (Table 4). Differentially regulated genes between *CTNNB1* mutant and non-mutant melanoma patients can be found in Appendix A. As expected, Enrichr analysis (https://maayanlab.cloud/Enrichr/ (accessed in February 2022)) of the 100 top differentially expressed genes revealed the Wnt-β Catenin Signaling pathway as the second pathway after fatty acid metabolism in the MSigDB Hallmark 2020 representing well-defined biological states or processes (Appendix A). In addition, the Wnt signaling pathway and pluripotency was the second pathway in BioPlanet 2019 and WikiPathway 2021 Human. Comparison of transcriptomic expression (measured in transcripts per million [TPM]) of multiple genes involved in this Hallmark WNT β-catenin signaling pathway (Appendix A) showed significant differences in the expression of *CTNNB1*-mutant and *CTNNB1*-wild type melanoma patients (Figure 3A). Expression of *AXIN2* (*p* ≤ 0.0001), *NKD1* (*p* = 0.003), *TP53* (*p* = 0.003), *HEY1* (*p* = 0.03), *PSEN2* (*p* = 0.001), and *CUL1* (*p* = 0.01) was significantly higher in *CTNNB1* mutant melanoma, supporting these mutations leading to an over activation of the Wnt pathway. Transcriptomic expression of *CTNNB1* was not elevated in *CTNNB1*-mutant melanoma. However, generally in the entire cohort, higher *CTNNB1* expression levels were correlated with expression of *AXIN1* (*p* = 0.009), *AXIN2* (*p* ≤ 0.001), *HEY1* (*p* ≤ 0.0001), *PSEN2* (*p* ≤ 0.0001), *PPARD* (*p* ≤ 0.0004), and *CUL1* (0.04) (Figure 3B). No significant difference in OS and PFS between *CTNNB1*-mutant and *CTNNB1*-wild type patients was observed (Figure 3C,D).

## 4. Discussion

*CTNNB1* mutations occur in both benign and malignant melanocytic tumors with a deep penetrating nevus-like phenotype. We identified *CTNNB1* mutations in two types of melanocytic tumors, benign nevi and malignant melanoma. Detailed histological, mutation, copy number and methylation analysis can clearly distinguish benign from malignant tumors. In addition, we investigated recurrent *CTNNB1* exon 3 mutations in the largest cohort reported to date to determine whether these mutations are associated with specific features relevant in clinical patient management.

Mutations of the β-catenin pathway have been reported to transform the phenotype of a *BRAF*-mutated common nevus into that of a deep penetrating nevus, including increased pigmentation, cell volume, and cyclin D1 levels in the nucleus [8]. Mutational activation of the MAP kinase and β-catenin pathways are practically pathognomic of the characteristic DPN phenotype. Data have also suggested that constitutive β-catenin pathway activation promotes tumorigenesis by overriding dependencies on the microenvironment that constrain proliferation of common nevi, with DPN-like melanoma harboring additional oncogenic mutations; further, these data identified DPN as an intermediate melanocytic neoplasm, positioned between benign nevus and DPN-like melanoma [8]. Our histopathologically challenging cases confirmed that in all seven cases, additional mutations in either *BRAF*, *NRAS*, *NF1* or further MAP kinase related genes were present. Interestingly, mutations in the *TERT* promoter region were only present in tumors identified as melanoma, and not in benign *CTNNB1*-mutant melanocytic tumors. Methylation profiling allowed a clear differentiation between benign and malignant (ID 3 + 6) *CTNNB1*-mutant tumors, underlining the potential of molecular and methylation analysis for further characterization of challenging cases.

Histopathologic evaluation remains the gold standard to classify melanocytic tumors and assess their likely clinical/biological potential. In most cases, including deep penetrating nevi, conventional histologic analysis is sufficient to distinguish benign from malignant tumors. However, in some histologically ambiguous tumors with deep penetrating morphology pathologic classification and determination of biological potential may not be clear-cut. In these difficult cases, genetic analysis may be a helpful additional tool in classifying deep penetrating tumors, as mutation profiles differ between primary melanomas and benign melanocytic tumors. The presented cases illustrate the potential diagnostic value of mutation profiling in a clinical setting.

Activating mutations (i.e., *BRAF, NRAS,* etc.) are found in both benign and malignant tumors, i.e., nevi and melanoma. A common theory is the acquisition of additional genetic events lead to tumors progressing, eventually tipping the balance from benign to malignant proliferations. Other potentially relevant events, such as DNA replication errors, have been discussed in odontogenic cysts and tumors which are also mainly benign despite harboring activating MAP Kinase or *CTNNB1* mutations [42].

To assess the role of *CTNNB1* mutations in advanced melanoma, we screened our large genetic melanoma database identifying 38 tumors—to our knowledge the largest cohort of *CTNNB1*-mutated advanced melanoma reported to date. As described previously, these tumors are rare [7,9]. Oulès et al., reported three NMM (30%), three SSM (30%), two lentigo malignant melanoma (LMM) (20%), one ALM (10%) and one deep-penetrating nevus-like melanoma (10%). Our cohort included 18% NMM, 13% SSM, 13% MUP, and 8% ALM, demonstrating a comparable distribution. In many cases, a specific melanoma subtype was not reported. However, our data and previous studies underline that melanocytic tumors harboring *CTNNB1* mutations often do not have a deep penetrating phenotype.

Comparing *CTNNB1* mutation frequencies with overall low mutation numbers is difficult, but the distribution we observed is comparable to previous reports. The most frequent *CTNNB1* mutations we observed were in S45, G34, T41 and S33 (found in 34%, 13%, 11% and 5% in our and 60%, 10%, 20% and 10% of cases in the Oulès et al., cohort, respectively) [7]. Concurrent mutations present were similar to results by Oulès et al., where an additional *BRAF* mutation was present in 55% of patients in our cohort (compared to 60% in [7]), but *NRAS* mutations were more frequent in our patients (34% compared to 20% in [7]). In our cohort, 4/38 patients had no additional mutations in *BRAF* or *NRAS* genes, showing that [9], *CTNNB1* mutations can occur in tumors not harboring these gene mutations [7]. Two of these four melanomas had mutations in *NF1*, one in *KIT* and one in *GNA11* (data not shown). A high rate of co-occurrence of MAPK-activating mutations (*BRAF/NRAS/NF1)* and *CTNNB1* mutations favors the hypothesis that mutations in *CTNNB1* display Wnt/β-catenin signaling proliferative hallmarks and cooperate with MAPK pathways [9].

For patients with advanced melanoma in whom *CTNNB1* mutations are unexpectedly discovered during routine molecular profiling, the extent to which they might impact patient therapy may be the most important question to consider. We found better therapeutic responses in patients receiving targeted therapy regimes (ORR 22%; DCR 44%, PR 22%) compared to immune checkpoint inhibition (ORR 11%; DCR 11%, PR 11%). These data are consistent with the previously described synergistic activity between Wnt/β-catenin signaling activation and BRAF inhibitors to reduce melanoma growth in vitro and in vivo [43]. Oncogenic signals have been postulated to mediate cancer immune evasion and resistance to immunotherapies [44]. Data has suggested active β-catenin signaling results in T cell exclusion and lack of T cell infiltrate driving resistance to anti-PD-L1 and anti-CTLA-4 immunotherapies [45]. Within this work, Spranger et al. identified the Wnt/β-catenin pathway as the first defined tumor-intrinsic oncogene pathway that can abort the induction of antitumor T cell responses, prevent the T cell-inflamed tumor microenvironment, and generate resistance to checkpoint blockade therapy [46]. Using transcriptomic data from a cohort of >700 melanoma patients (primaries and metastasis), Nsengimana et al. could show that low-immune/β-catenin high expressing tumor patients show fewer pathologist-reported brisk tumor infiltrating lymphocytes (TILs) and significantly worsened melanoma-specific survival, underlining oncogenic potential of the Wnt pathway [47]. Significant changes on the transcriptomic level of genes involved in the Wnt/β-catenin signaling pathway underline a biological regulation of this pathway in *CTNNB1*-mutant melanoma. The ORR of 11% in our cohort would support *CTNNB1*-mutant tumors responding poorly to immune checkpoint therapy. However, performing an additional analysis of outcomes of *CTNNB1*-mutant and *CTNNB1*-wild type melanoma patients from a recently published anti-PD1 monotherapy-treated melanoma cohort [23] an ORR of 67% (DCR 78%) in *CTNNB1*-mutant compared to 36% (DCR 50%) in *CTNNB1*-wild type melanoma patients, was observed, arguing immune checkpoint inhibition therapy can be effective for patients with *CTNNB1*-mutant tumors. However, as the number of treated patients in both our and the Liu et al. study remain limited, larger studies are required.

Based on our and existing data to date, we believe no clear-cut recommendation concerning therapeutic approach or prognosis concerning survival can be made for *CTNNB1*-mutant melanoma. Both types of therapy, targeted and immune-checkpoint inhibition, have shown efficacy. Larger studies, optimally in a prospective fashion will be required to further elucidate if *CTNNB1* mutation status should be clearly linked to specific systemic therapy recommendations in advanced melanoma patients. A limitation of the study is the low number of patients. Studies with larger numbers of difficult to classify melanocytic lesions as well as CTNNB1 mutated melanomas may offer further insights. Considering performing both NGS sequencing and methylation arrays is cost intensive and not universally available, larger cohort studies may also help identify a selection of relevant gene mutation and methylation sites enabling a more focused cost-effective analysis.

In summary, we report the largest cohort of *CTNNB1*-mutated melanocytic tumors identified so far. *CTNNB1* mutations can be found in difficult-to-classify tumors with deep penetrating morphology and additional molecular and methylation analysis can help differentiate between benign and malignant tumors in these cases. *CTNNB1*-mutant melanomas were found to originate from different locations and only rarely demonstrated a deep penetrating phenotype. Therapeutic responses to both targeted and ICI therapy were observed.

## 5. Conclusions

-Mutation analysis in conjunction with methylation analysis can be a diagnostic aid in determining the dignity in some cases of deep penetrating melanocytic tumors-*CTNNB1*-mutant melanoma comprises ~1–2% of melanoma-Histologic characteristics can show a deep penetrating nevus, but can also be any melanoma subtype

## Figures and Tables

**Figure 1 cancers-14-04066-f001:**
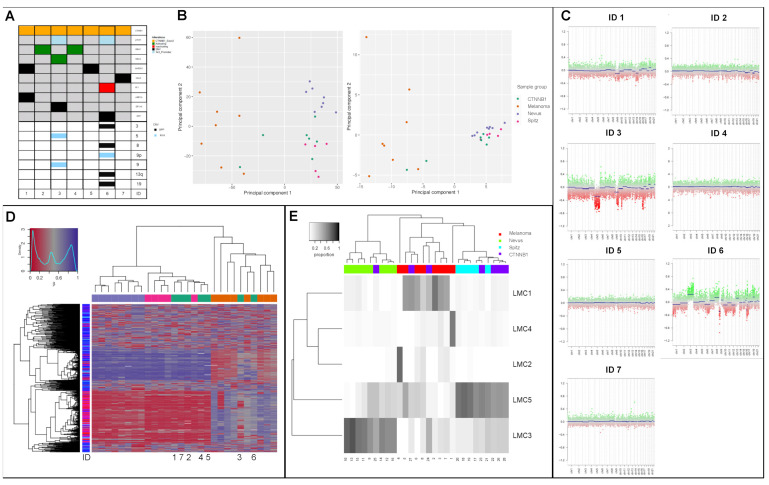
Oncoprint of all (*n* = 7) unclassified melanocytic lesions including CNV profiles (**A**). Principal component analysis (PCA) of all subgroups for all sites and genes (**B**). Percentage of total variance was 43.0% for PCA 1 and 8.2% for PCA 2 or genes and 33.5% for PCA 1 and 9.1% for PCA2 for sites. Predicted CNV profiles from all seven patients with *CTNNB1*-mutation by ID. Heatmap showing average DNA methylation of 10,000 most differentially methylated sites. Hierarchical clustering is based on a correlation distance and complete linkage (**C**). Heatmap showing proportions of the LMCs in all patient samples ((**D**), rows: *n* = 8 nevi, *n* = 8 malignant melanoma, *n* = 5 Spitz nevi, *n* = 7 *CTNNB1*-mutant melanocytic lesions) (**E**). Lambda values of 0.01 were used. LMC, latent methylation components.

**Figure 2 cancers-14-04066-f002:**
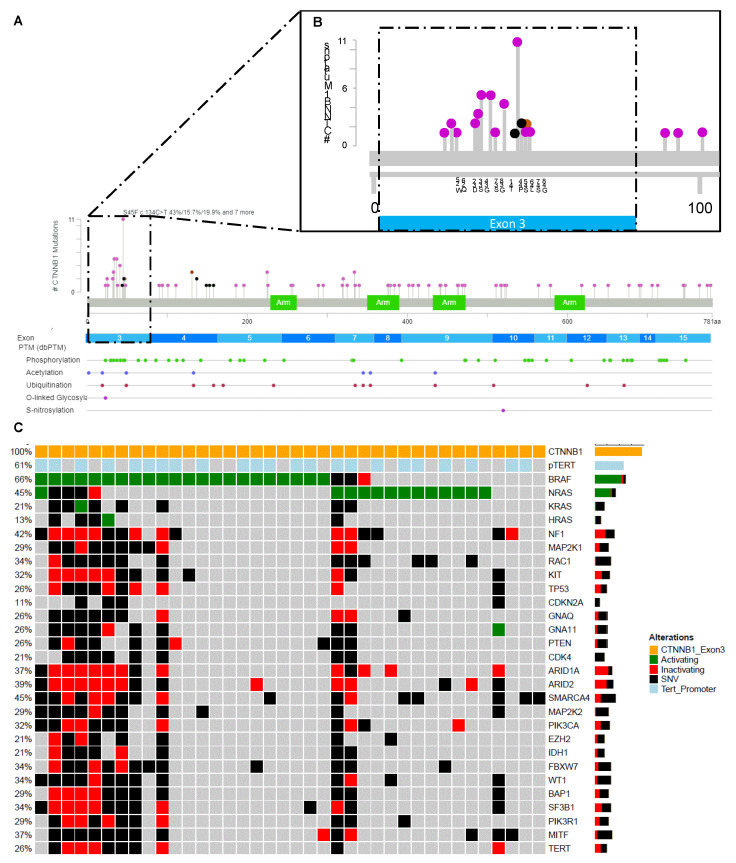
Figure was generated and adapted using the MutationMapper provided by www.sbioportal.org [40,41]. All identified mutations of melanoma harboring a *CTNNB1* mutation in the hotspot region in exon 3 of *CTNNB1* and additional mutations within these tumors were mapped (**A**). Hotspot region exon 3 is shown in detail in (**B**). Oncoprint of all *n* = 38 *CTNNB1* mutant melanoma samples (**C**).

**Figure 3 cancers-14-04066-f003:**
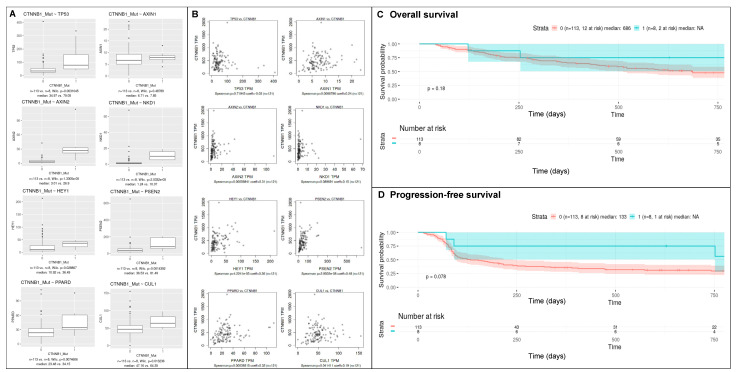
Transcriptomic expression of hallmark Wnt/β-catenin signaling pathway genes in *CTNNB1*-mutant versus non-mutant melanoma patients treated with anti-PD1 monotherapy (**A**). Gene expression was measured in transcripts per million (TPM). Wilcoxon sum rank test was used for comparison. Spearman correlation of transcriptomic gene expression of *CTNNB1* (y-axis) and hallmark Wnt/β-catenin signaling pathway genes (**B**). Overall (**C**) and progression-free survival (**D**) in *CTNNB1*-mutant compared to non-mutant melanoma patients treated with anti-PD1 monotherapy. Survival was measured in days.

**Table 1 cancers-14-04066-t001:** Clinical and genetic characteristics.

ID	Sex	Age *	Locali-Sation	Diagnosis	BRAF	NRAS	Tert Promoter	Other	Wnt Pathway	PD-L1 Staining	CNV Gains	CNV Losses
1	m	12	Lower extremity	Spitz nevus **	wt	wt	wt	MAP2K1_I103_K104del M DACH1_G82_G83del	CTNNB1 S37F			
2	m	11	NA	Deep penetrating blue nevus	V600E	wt	wt		CTNNB1 S37F, CTNNB1 N287S			
3	m	59	Head	Deep penetrating nevus like Melanoma	wt	Q61R	hTERTP 1,295,250 G>A	EIF1AX W70S	CTNNB1 S45P	n.d.		5, 9
4	f	45	NA	Combined Nevus	V600E	wt	wt		CTNNB1 S45F	na		
5	m	47	Trunk	Deep-penetrating nevus	wt	wt	wt	MAP2K1 I103_K104del	CTNNB1 S37F	negative (<1% of tumor cells)		
6	m	64	Back	Malignant melanoma under the picture of a deeply penetrating blue nevus	wt	wt	hTERTP 1,295,228 G>A	NF1 R1241 *		positive (5% of tumor cells)	3, 8, 13q, 19	9p
7	m	4	Head	Congenital nevus cell nevus	wt	wt		HRAS M72delins	CTNNB1 S33F	na		

Abbreviations: f, female; m, male; MM, malignant melanoma; na, not available; n.d. not determinable; PD-L1, programmed death ligand-1; wt, wildtype; mut, mutated. * at diagnosis ** mainly intradermal.

**Table 2 cancers-14-04066-t002:** Clinical characteristic of melanoma patients with CTNNB1 mutation.

	CTNNB 1 Mutant Melanoma Patients (*N* = 38)
SEX	
MALE	27 (71.1%)
FEMALE	11 (28.9%)
AGE AT DIAGNOSIS	
MEDIAN [MIN, MAX]	59.0 [39.0, 90.0]
HISTOLOGICAL SUBTYPE	
MM (UNSPECIFIED SUBTYPE)	17 (44.7%)
NMM	7 (18.4%)
SSM	5 (13.2%)
MUP	5 (13.2%)
ALM	3 (7.9%)
SPITZOID MM	1 (2.6%)
LOCALISATION	
LOWER EXTREMITY	14 (36.8%)
TRUNK	12 (31.6%)
HEAD/NECK	5 (13.2%)
MUP	4 (10.5%)
UPPER EXTREMITY	2 (5.3%)
UNKNOWN	1 (2.6%)
TUMOR THICKNESS (MM)	
MEDIAN [MIN, MAX]	2.55 [0.42, 14.5]
UNKNOWN	6 (15.8%)
ULCERATION	
YES	18 (47.4%)
NO	11 (28.9%)
UNKNOWN	9 (23.7%)
STAGE AT DIAGNOSIS	
IA	1 (2.6%)
IB	4 (10.5%)
IIA	4 (10.5%)
IIB	3 (7.9%)
IIC	1 (2.6%)
IIIA	3 (7.9%)
IIIB	5 (13.2%)
IIIC	4 (10.5%)
IV	4 (10.5%)
UNKNOWN	9 (23.7%)
LYMPH NODE METASTASIS	
NO	13 (34.2%)
YES	25 (65.8%)
LUNG METASTASIS	
NO	26 (68.4%)
YES	12 (31.6%)
LIVER METASTASIS	
NO	26 (68.4%)
YES	11 (28.9%)
UNKNOWN	1 (2.6%)
BONE METASTASIS	
NO	34 (89.5%)
YES	4 (10.5%)
CNS METASTASIS	
NO	27 (71.1%)
YES	11 (28.9%)
OTHER METASTASIS	
NO	14 (36.8%)
YES	24 (63.2%)
OS (DAYS FROM DIAGNOSIS)	
MEAN (SD)	2480 (1790)
MEDIAN [MIN, MAX]	2060 [59.0, 6310]
SURVIVAL STATUS	
ALIVE	24 (63.2%)
DECEASED	14 (36.8%)
STAGE AT DATA CUT	
IIIA	1
IIIB	4
IIIC	6
IV	25
UNKNOWN	2
FIRST SYSTEMIC THERAPY (IN ADVANCED DISEASE)	
ICI	9 (1 PR, 8 PD)
TT	9 (2 PR, 2 SD, 3 PD, 2 unknown)
OTHER	Chemo: 1 (PD) NIPAWILMA: 1 (PD) TriN 2755: 1 (SD)

Abbreviations: ALM, acrolentiginous melanoma; CNS, central nervous system; ICI, immune checkpoint inhibition; MM, malignant melanoma; MUP, melanoma of unknown primary; NMM, nodular malignant melanoma; OS, overall survival; PR, partial response; PD, progressive disease; TT, targeted therapy; SD, stable disease; SSM, superficial spreading melanoma.

**Table 3 cancers-14-04066-t003:** Mutations in CTNNB1 mutated melanoma.

CTNNB1 Mutation	Number (%) (*N* = 38)
S45	13 (34.2%)
G34	5 (13.2%)
S37	5 (13.2%)
T41	4 (10.5%)
S33	2 (5.3%)
W25	2 (5.3%)
D32	1 (2.6%)
G38	1 (2.6%)
G48	1 (2.6%)
L46	1 (2.6%)
P44	1 (2.6%)
Q26	1 (2.6%)
S47	1 (2.6%)
BRAF MUTATION	
V207E	18 (47.4%)
V207R	2 (5.2%)
W57*	1 (2.6%)
G73R	1 (2.6%)
P177FS	1 (2.6%)
P262L	1 (2.6%)
V207D	1 (2.6%)
WT	13 (34.2%)
NRAS MUTATION	
Q61K	5 (13.2%)
Q61R	3 (7.9%)
Q61L	2 (5.3%)
Q61H	2 (5.2%)
E132K	1 (2.6%)
G115R	1 (2.6%)
Q61V	1 (2.6%)
V152I	1 (2.6%)
V188M	1 (2.6%)
WT	21 (55.3%)
TERTP250 MUTATION	
YES	14 (36.8%)
TERTP228 MUTATION	
YES	11 (28.9%)
TERTP242 MUTATION	
YES	4 (10.5%)

**Table 4 cancers-14-04066-t004:** Clinical and transcriptomic characteristics of anti-PD-1 monotherapy treated melanoma patients from [23].

	CTNNB1 Wt (*N* = 135)	CTNNB1 Mut (*N* = 9)	*p*-Value	Overall (*N* = 144)
GENDER			0.31	
MALE	77 (57.0%)	2 (22.2%)		84 (58.3%)
FEMALE	58 (43.0%)	7 (77.8%)		60 (41.7%)
PRIMARY TYPE			0.29	
SKIN	99 (73.3%)	6 (66.7%)		105 (72.9%)
OCCULT	18 (13.3%)	1 (11.1%)		19 (13.2%)
MUCOSAL	10 (7.4%)	0 (0%)		10 (6.9%)
ACRAL	8 (5.9%)	2 (22.2%)		10 (6.9%)
STAGE			0.37	
IIIC	10 (7.4%)	0 (0%)		10 (6.9%)
M1A	7 (5.2%)	1 (11.1%)		8 (5.6%)
M1B	16 (11.9%)	2 (22.2%)		18 (12.5%)
M1C	102 (75.6%)	6 (66.7%)		108 (75.0%)
LDH ELEVATED			0.75	
NO	65 (48.1%)	5 (55.6%)		70 (48.6%)
YES	67 (49.6%)	4 (44.4%)		71 (49.3%)
UNKNOWN	3 (2.2%)	0 (0%)		3 (2.1%)
BRAIN METS			0.60	
NO	119 (88.1%)	9 (100%)		128 (88.9%)
YES	16 (11.9%)	0 (0%)		16 (11.1%)
CUT/SUBCUT METS			0.32	
NO	52 (38.5%)	5 (55.6%)		57 (39.6%)
YES	83 (61.5%)	4 (44.4%)		87 (60.4%)
LN METS			0.50	
NO	48 (35.6%)	2 (22.2%)		50 (34.7%)
YES	87 (64.4%)	7 (77.8%)		94 (65.3%)
LUNG METS			0.74	
NO	53 (39.3%)	4 (44.4%)		57 (39.6%)
YES	82 (60.7%)	5 (55.6%)		87 (60.4%)
LIVER/VISC METS			0.73	
NO	76 (56.3%)	6 (66.7%)		82 (56.9%)
YES	59 (43.7%)	3 (33.3%)		62 (43.1%)
BONE METS			0.42	
NO	105 (77.8%)	6 (66.7%)		111 (77.1%)
YES	30 (22.2%)	3 (33.3%)		33 (22.9%)
PROGRESSED			0.12	
NO	36 (26.7%)	5 (55.6%)		41 (28.5%)
YES	99 (73.3%)	4 (44.4%)		103 (71.5%)
DEAD			0.17	
NO	65 (48.1%)	7 (77.8%)		72 (50.0%)
YES	70 (51.9%)	2 (22.2%)		72 (50.0%)
BR			0.39	
CR	15 (11.1%)	2 (22.2%)		17 (11.8%)
PR	34 (25.2%)	4 (44.4%)		38 (26.4%)
MR	4 (3.0%)	0 (0%)		4 (2.8%)
SD	19 (14.1%)	1 (11.1%)		20 (13.9%)
PD	63 (46.7%)	2 (22.2%)		65 (45.1%)
PURITY			0.61	
MEAN (SD)	0.623 (0.238)	0.660 (0.249)		0.625 (0.238)
MEDIAN [MIN, MAX]	0.670 [0.100, 0.950]	0.700 [0.150, 0.930]		0.670 [0.100, 0.950]
TOTAL_MUTS			0.40	
MEAN (SD)	823 (1530)	2550 (4910)		931 (1930)
MEDIAN [MIN, MAX]	370 [13.0, 9590]	491 [21.0, 15,300]		380 [13.0, 15,300]
NONSYN_MUTS			0.37	
MEAN (SD)	542 (994)	1660 (3160)		612 (1250)
MEDIAN [MIN, MAX]	245 [10.0, 6250]	328 [17.0, 9840]		251 [10.0, 9840]
CLONAL_MUTS			0.37	
MEAN (SD)	412 (725)	1430 (2870)		475 (1010)
MEDIAN [MIN, MAX]	188 [6.00, 5400]	273 [14.0, 8920]		191 [6.00, 8920]
SUBCLONAL_MUTS			0.22	
MEAN (SD)	109 (467)	185 (243)		114 (456)
MEDIAN [MIN, MAX]	37.0 [2.00, 5230]	46.0 [3.00, 662]		37.5 [2.00, 5230]
HETEROGENEITY			0.72	
MEAN (SD)	0.199 (0.135)	0.207 (0.121)		0.199 (0.134)
MEDIAN [MIN, MAX]	0.174 [0.0202, 0.971]	0.176 [0.0691, 0.469]		0.174 [0.0202, 0.971]
TOTAL_NEOANTIGENS			0.42	
MEAN (SD)	1580 (2830)	4390 (7920)		1760 (3390)
MEDIAN [MIN, MAX]	726 [20.0, 18,500]	1090 [42.0, 24,600]		729 [20.0, 24,600]
CNA_PROP			0.03	
MEAN (SD)	0.199 (0.124)	0.111 (0.0854)		0.193 (0.123)
MEDIAN [MIN, MAX]	0.181 [0.00670, 0.902]	0.0791 [0.0104, 0.236]		0.178 [0.00670, 0.902]

## Data Availability

The raw Illumina HumanMethylationEPIC BeadChip array data generated in this study are available in EGA upon acceptance. Other data that support the findings of this study are available from the corresponding author upon request.

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
