# Peer review of "Genetic and Methylation Analysis of CTNNB1 in Benign and Malignant Melanocytic Lesions"

_cancers, 2022, doi:10.3390/cancers14174066_

Round 1
Reviewer 1 Report
The authors of the study titled: “Genetic and methylation analysis of CTNNB1 in benign and malignant melanocytic lesions” has analyzed a cohort of seven melanocytic tumor with CTNNB1 mutations and compared them to a control group, which consisted of eight benign nevi, eight malignant melanoma and 5 spitz nevi. Additionally, they analyzed clinical and mutational data from 38 CTNNB1 mutant melanoma samples. Moreover, the authors investigated publicly available melanoma dataset (Liu et al.). Generally, the analyses performed in this paper could potentially be interesting however there are several major issues I recognized in herein presented study.
1. Unfortunately, the poor quality of the figures impedes proper review of this paper (e.g. Fig 1a, Fig. 3).
2. Some information is missing:
a. Cohort information of the control group used for comparisons with the seven CTNNB1-mutant melanocytic lesions.
b. Distinct method section of the analyses of the public melanoma dataset (Liu et. al). For example, there is no information on how the survival analyses were performed.
1. The cohort of the difficult to classify melanocytic lesions is extremely small and moreover is comprised only with one female patient which makes the cohort biased.
2. The authors claim that mutational and methylation screening could potentially help to classify some CTNNB1 mutant melanocytic lesions. However, it is not cost effective to perform agnostic methylation and mutational analyses in the clinic. Could the authors identify specific methylations/mutations that could be detected in targeted way?
3. Why the authors did not analyze the TCGA melanoma dataset which also contains mutational and methylation data?
Author Response
Response to Reviewer 1 Comments
The authors of the study titled: “Genetic and methylation analysis of CTNNB1 in benign and malignant melanocytic lesions” has analyzed a cohort of seven melanocytic tumor with CTNNB1 mutations and compared them to a control group, which consisted of eight benign nevi, eight malignant melanoma and 5 spitz nevi. Additionally, they analyzed clinical and mutational data from 38 CTNNB1 mutant melanoma samples. Moreover, the authors investigated publicly available melanoma dataset (Liu et al.). Generally, the analyses performed in this paper could potentially be interesting however there are several major issues I recognized in herein presented study.
Point 1: Unfortunately, the poor quality of the figures impedes proper review of this paper (e.g. Fig 1a, Fig. 3).
Response 1: We would like to thank Reviewer 1 for noticing this important point. We adapted Figures 1-3 aiming to impriove the quality of each figure and uploaded these new versions. In addition, we have omitted the confidence intervals for the survival curves in order to simplify the presentation.
Point 2: Some information is missing:
- Cohort information of the control group used for comparisons with the seven CTNNB1-mutant melanocytic lesions.
- Distinct method section of the analyses of the public melanoma dataset (Liu et. al). For example, there is no information on how the survival analyses were performed.
Response 2:
- We thank Reviewer 1 for raising this important point. We have adapted the following sentences into the methods section on page (p.) 3 and included a new supplementary table 1:
“Clinical characteristics of patients from the control group are shown in supplementary table 1 (II). Additional information regarding these cases can be found in a previous manuscript [1].”
- We thank Reviewer 1 for this valid comment, we have included the following part into the methods section on p. 3:
“For survival analysis, progression-free survival (PFS) and overall survival (OS) were defined as time from therapy start until disease progression or death, respectively; if no such event occurred, the date of the last patient contact was used as endpoint of survival assessment (censored observation).”
Point 3: The cohort of the difficult to classify melanocytic lesions is extremely small and moreover is comprised only with one female patient which makes the cohort biased.
Response 3: We agree with Reviewer 1 that a larger cohort of difficult to classify melanocytic lesions would be desirable. As stated by us and others [2, 3] melanoma harboring CTNNB1 mutations are rare and often mutations within the β-catenin/Wnt pathway are not screened for lacking therapeutical implications to date. Even rarer are histologically difficult to classify melanocytic lesions known to have CTNNB1 mutations. The collection we present within this manuscript was collected over a time period of eight years in one of the largest skin cancer centers in Germany. Biases such as the gender bias the reviewer identified can occur within small cohorts. Therefore, one of our goals with this manuscript is to raise awareness of this subset of benign and malignant CTNNB1-mutated melanocytic lesions and to highlight future clinical implications. Compared to previous studies on CTNNB1 mutant melanoma, our cohort with n=38 advanced melanoma is large [2, 3]. To underline this limitation within our study, we included following sentence into the discussion section on p. 18:
“A limitation of the study is the low number of patients. Studies with larger numbers of difficult to classify melanocytic lesions as well as CTNNB1 mutated melanomas may offer further insights.”
Point 4: The authors claim that mutational and methylation screening could potentially help to classify some CTNNB1 mutant melanocytic lesions. However, it is not cost effective to perform agnostic methylation and mutational analyses in the clinic. Could the authors identify specific methylations/mutations that could be detected in targeted way?
Response 4: This is certainly a justified concern. Many centers lack any kind of genetic analysis facility and having both a Next-Generation-Sequencing and methylation array facility will be restricted to larger institutions. Additionally cost concerns may be relevant. These are rare tumors however and in such cases sending them for further analysis to a fully equiped facility is arguably worth the extra time and cost for affected patients. Currently, we can´t offer specific methylation sites or mutation sites allowing a more targeted analysis approach, potentially larger future studies can identify such targets. We have added this point to the limitation part of the discussion section. It now reads:
“Considering performing both NGS sequencing and methylation arrays is cost intensive and not universally available, larger cohort studies may help identify a selection of relevant gene mutation and methylation sites enabling a more focused cost-effective analysis.”
Point 5: Why the authors did not analyze the TCGA melanoma dataset which also contains mutational and methylation data?
Response 5: We thank Reviewer 1 for raising this point. We indeed used the TCGA dataset to idetify mutations within the Skin Cutaneous Melanoma (Firehose Legacy) cohort and saw n=21 mutations within CTNNB1 in n=470 patients/n=473 samples (Table 1). Hereof, n=12 were located within the hotspot region of exon 3 (Table 1, marked in bold), adding up to a total of 2.6% (Figure 1).
As we were interested mainly in the potential therapeutical value of CTNNB1 mutations in the context of immune checkpoint inhibitor therapy, this data was of no further interest for our study and we decided to use the Liu et al. dataset including advanced melanoma patients receiving anti-PD1 monotherapy instead [4].
Figure 1
Generated using www.cbioportal.org [5]
Table 1: Characteristics of CTNNB1 mutant melanoma from the Skin Cutaneous Melanoma cohort (TCGA, Firehose Legacy)
|
Study of Origin |
Sample ID |
Cancer Type Detailed |
Protein Change |
Mutation Type |
Variant Type |
Start Pos |
End Pos |
Ref |
Var |
|
Skin Cutaneous Melanoma (TCGA, Firehose Legacy) |
TCGA-D9-A6EG-06 |
Cutaneous Melanoma |
S45P |
Missense_Mutation |
SNP |
41266136 |
41266136 |
T |
C |
|
Skin Cutaneous Melanoma (TCGA, Firehose Legacy) |
TCGA-EB-A42Y-01 |
Cutaneous Melanoma |
S45F |
Missense_Mutation |
SNP |
41266137 |
41266137 |
C |
T |
|
Skin Cutaneous Melanoma (TCGA, Firehose Legacy) |
TCGA-GN-A269-01 |
Melanoma |
G34E |
Missense_Mutation |
SNP |
41266104 |
41266104 |
G |
A |
|
Skin Cutaneous Melanoma (TCGA, Firehose Legacy) |
TCGA-ER-A19O-06 |
Cutaneous Melanoma |
T41A |
Missense_Mutation |
SNP |
41266124 |
41266124 |
A |
G |
|
Skin Cutaneous Melanoma (TCGA, Firehose Legacy) |
TCGA-FS-A4F9-06 |
Cutaneous Melanoma |
G34R |
Missense_Mutation |
SNP |
41266103 |
41266103 |
G |
A |
|
Skin Cutaneous Melanoma (TCGA, Firehose Legacy) |
TCGA-EE-A2M6-06 |
Cutaneous Melanoma |
S45del |
In_Frame_Del |
DEL |
41266134 |
41266136 |
CTT |
- |
|
Skin Cutaneous Melanoma (TCGA, Firehose Legacy) |
TCGA-FS-A1YY-06 |
Cutaneous Melanoma |
S45C |
Missense_Mutation |
SNP |
41266137 |
41266137 |
C |
G |
|
Skin Cutaneous Melanoma (TCGA, Firehose Legacy) |
TCGA-D9-A149-06 |
Cutaneous Melanoma |
T41I |
Missense_Mutation |
SNP |
41266125 |
41266125 |
C |
T |
|
Skin Cutaneous Melanoma (TCGA, Firehose Legacy) |
TCGA-EB-A42Y-01 |
Cutaneous Melanoma |
T41I |
Missense_Mutation |
SNP |
41266125 |
41266125 |
C |
T |
|
Skin Cutaneous Melanoma (TCGA, Firehose Legacy) |
TCGA-EB-A3Y7-01 |
Cutaneous Melanoma |
D32G |
Missense_Mutation |
SNP |
41266098 |
41266098 |
A |
G |
|
Skin Cutaneous Melanoma (TCGA, Firehose Legacy) |
TCGA-EE-A180-06 |
Cutaneous Melanoma |
K292E |
Missense_Mutation |
SNP |
41267290 |
41267290 |
A |
G |
|
Skin Cutaneous Melanoma (TCGA, Firehose Legacy) |
TCGA-EB-A5SH-06 |
Cutaneous Melanoma |
T41N |
Missense_Mutation |
SNP |
41266125 |
41266125 |
C |
A |
|
Skin Cutaneous Melanoma (TCGA, Firehose Legacy) |
TCGA-D3-A1Q5-06 |
Cutaneous Melanoma |
C429G |
Missense_Mutation |
SNP |
41275119 |
41275119 |
T |
G |
|
Skin Cutaneous Melanoma (TCGA, Firehose Legacy) |
TCGA-EE-A181-06 |
Cutaneous Melanoma |
P714L |
Missense_Mutation |
SNP |
41280628 |
41280628 |
C |
T |
|
Skin Cutaneous Melanoma (TCGA, Firehose Legacy) |
TCGA-D3-A2JN-06 |
Cutaneous Melanoma |
P687S |
Missense_Mutation |
SNP |
41278183 |
41278183 |
C |
T |
|
Skin Cutaneous Melanoma (TCGA, Firehose Legacy) |
TCGA-D3-A2JN-06 |
Cutaneous Melanoma |
P687L |
Missense_Mutation |
SNP |
41278184 |
41278184 |
C |
T |
|
Skin Cutaneous Melanoma (TCGA, Firehose Legacy) |
TCGA-EE-A3AC-06 |
Cutaneous Melanoma |
Q545* |
Nonsense_Mutation |
SNP |
41275738 |
41275738 |
C |
T |
|
Skin Cutaneous Melanoma (TCGA, Firehose Legacy) |
TCGA-D9-A6E9-06 |
Cutaneous Melanoma |
D390E |
Missense_Mutation |
SNP |
41274920 |
41274920 |
T |
G |
|
Skin Cutaneous Melanoma (TCGA, Firehose Legacy) |
TCGA-D9-A6EC-06 |
Cutaneous Melanoma |
L762P |
Missense_Mutation |
SNP |
41280772 |
41280772 |
T |
C |
|
Skin Cutaneous Melanoma (TCGA, Firehose Legacy) |
TCGA-EE-A182-06 |
Cutaneous Melanoma |
W25* |
Nonsense_Mutation |
SNP |
41266077 |
41266077 |
G |
A |
|
Skin Cutaneous Melanoma (TCGA, Firehose Legacy) |
TCGA-EE-A29A-06 |
Cutaneous Melanoma |
P606L |
Missense_Mutation |
SNP |
41277853 |
41277853 |
C |
T |

Reviewer 2 Report
Your study shows the improved response rate of CTNNB1-mutant melanoma to anti-PD-1 compared to CTNNB1-wild type tumors and may demonstrate immune checkpoint inhibition is an effective treatment in patients with CTNNB1-mutant melanoma. Larger future studies will be informative.
Author Response
Response to Reviewer 2 Comments
Your study shows the improved response rate of CTNNB1-mutant melanoma to anti-PD-1 compared to CTNNB1-wild type tumors and may demonstrate immune checkpoint inhibition is an effective treatment in patients with CTNNB1-mutant melanoma. Larger future studies will be informative.
Response 1: We appreciate the feedback obtained from Reviewer 2 and would like to thank him for his positive response. We agree that larger future studies in this context will be informative.

Reviewer 3 Report
This is a concise description of a small retrospective study of 7 CTNNB1 mutant patients with histopathologically difficult cases of melanocytic tumors from a centre in Germany. The authors show, that in difficult cases of deep penetrating tumors, genetic analysis may be a helpful additional tool in classifying the dignity, as mutation profiles differ between primary melanomas and benign melanocytic tumors.
These 7 cases illustrate the potential diagnostic value of mutation profiling in a clinical setting. As a result, malignant CTNNB1-mutant melanomas demonstrated a different genetic profile, grouping clearly with other non-CTNNB1 melanoma in methylation assays.
On the whole, this article is well written. The use of English is excellent.
However, some issues need to be clarified.
Material and Methods: The question is why were the seven cases difficult to diagnose. The authors should demonstrate the hallmarks of histologically ambiguous findings as an indication for molecular pathology.
Patient selection: Start with a clear statement that you address 3 topics, i.e. "a cohort of seven" CTNNB1 mutant patients for diagnostics (page 3) vs database of 38 CTNNB1 mutant melanoma patients for clinical characteristics (chapter 3.3) and another database of 144 melanoma patients with or without CTNNB1 mutations treated with anti-PD-1 monotherapy (chapter 3.5).
Table 1: Explain "na" and apply the introduced abbreviation for "n.d." in the table.
Discussion: The quoted article by Yeh et al (8) identifying DPN as an intermediate melanocytic neoplasm, positioned between benign nevus and DPN-like melanoma, needs further explanation. The seven cases in this study with challenging histopathology presented additional mutations in BRAF, NRAS, NF1 or further MAP kinase related genes. This additional molecular information, previously considered to be the hallmark drivers of cancer, does so far not explain the biological potential or risk for malignant transformation but may be an epiphenomenon since various kind of benign melanocytic (as well as non-melanocytic) skin tumors may harbor oncogenic mutations but without any risk of malignant transformation, i.e. blue nevi, epitheloid and spindle cell Spitz nevi, congenital nevi, dysplastic nevi. Despite the presence of such alterations, the vast majority of these lesions remain benign and never progress to the stage of malignant transformation. Therefore, the question arises, whether they do result from DNA replication errors, as discussed by Diniz MG, Gomes CC, de Sousa SF, Xavier GM, Gomez RS: Oncogenic signalling pathways in benign odontogenic cysts and tumours (Oral Oncol. 2017; 72:165-173).
Please, clarify and explain the term “advanced” melanoma in the context of the role of CTNNB1 mutations in advanced melanoma. Does that imply a difference to non-advanced melanoma?
The authors should avoid inappropriate self-citations (i.e. 1,2,12,14,15,19,20,22,23,24,25,26).
Did you detect inappropriate self-citations by authors
The statistical analysis appears to be sound, the chosen test types are appropriate to the data set. The graphical representations of the results are a useful aid to understanding the text and clearly demonstrate the described trends. The references are from a wide variety of sources and I found no instances of self-citation.
In summary, this short article could be of value and is, on the whole, well written. However, several small but significant improvements could, and should, be made.
Author Response
Response to Reviewer 3 Comments
This is a concise description of a small retrospective study of 7 CTNNB1 mutant patients with histopathologically difficult cases of melanocytic tumors from a centre in Germany. The authors show, that in difficult cases of deep penetrating tumors, genetic analysis may be a helpful additional tool in classifying the dignity, as mutation profiles differ between primary melanomas and benign melanocytic tumors.
These 7 cases illustrate the potential diagnostic value of mutation profiling in a clinical setting. As a result, malignant CTNNB1-mutant melanomas demonstrated a different genetic profile, grouping clearly with other non-CTNNB1 melanoma in methylation assays.
On the whole, this article is well written. The use of English is excellent.
However, some issues need to be clarified.
Point 1: Material and Methods: The question is why were the seven cases difficult to diagnose. The authors should demonstrate the hallmarks of histologically ambiguous findings as an indication for molecular pathology.
Response 1: We have added a description of this to the Material and Methods section, it now reads:
“The seven difficult to classify cases were either seen at our department or referred to our department from other institutions for further analysis. Tumors with deep-penetration nevus-like morphology characteristically show large cells showing no maturation toward deeper tissue and an infiltrative growth pattern, expanding in an interstitial fashion into the tissue. These are traits than can also be seen in melanoma. In immunohistochemistry, DPN-like tumors can express HMB45 at certain amounts and demonstrate some level of reactivity to MIB (or Ki-67). All cases deemed not clearly benign by conventional histopathological analysis were passed on for genetic analysis.”
Point 2: Patient selection: Start with a clear statement that you address 3 topics, i.e. "a cohort of seven" CTNNB1 mutant patients for diagnostics (page 3) vs database of 38 CTNNB1 mutant melanoma patients for clinical characteristics (chapter 3.3) and another database of 144 melanoma patients with or without CTNNB1 mutations treated with anti-PD-1 monotherapy (chapter 3.5).
Response 2: We would like to thank Reviewer 3 for raising this important point and included following statements to underline the different cohorts used within this study on p. 3:
“To address all aims, three distinct cohorts were studied. (I) A cohort of seven CTNNB1-mutant melanocytic tumors was analysed using genome-wide DNA methylation in conjunction with copy number variation (CNV) and mutation profiling (chapter 3.1 and 3.2). The control group consisted of eight benign nevi, eight malignant melanoma, and five Spitz nevi. Clinical characteristics of patients from the control group are shown in supplementary table 1 (II). Additional information regarding these cases can be found in a previous manuscript [1]. (II) Clinical and mutational data from n=38 CTNNB1-mutant melanoma patients was retrieved from routinely performed NGS melanoma panel analysis and the medical databases/documentation system of the University Hospital Essen (chapter 3.3 and 3.4). Only melanoma patients with metastatic disease (advanced melanoma) and therefore clinically confirmed malignant disease in stages IIIA or higher were included (see also Table 1). (III) Anti-PD1-treated melanoma patients were retrieved from [4]. Chi square and Fisher t tests were used for comparison of categorical variables as applicable. A Kruskal-Wallis test was used for continuous variables. Wilcoxon rank sum tests were used for continuous variables in R 4.2.0 (chapter 3.5).”
Point 3: Table 1: Explain "na" and apply the introduced abbreviation for "n.d." in the table.
Response 3: We thank Reviewer 3 for noticing this missing explanation. We included the explanation for ‘na’ (not available) into the abbreviation part of Table 1.
Point 4: Discussion: The quoted article by Yeh et al (8) identifying DPN as an intermediate melanocytic neoplasm, positioned between benign nevus and DPN-like melanoma, needs further explanation. The seven cases in this study with challenging histopathology presented additional mutations in BRAF, NRAS, NF1 or further MAP kinase related genes. This additional molecular information, previously considered to be the hallmark drivers of cancer, does so far not explain the biological potential or risk for malignant transformation but may be an epiphenomenon since various kind of benign melanocytic (as well as non-melanocytic) skin tumors may harbor oncogenic mutations but without any risk of malignant transformation, i.e. blue nevi, epitheloid and spindle cell Spitz nevi, congenital nevi, dysplastic nevi. Despite the presence of such alterations, the vast majority of these lesions remain benign and never progress to the stage of malignant transformation. Therefore, the question arises, whether they do result from DNA replication errors, as discussed by Diniz MG, Gomes CC, de Sousa SF, Xavier GM, Gomez RS: Oncogenic signalling pathways in benign odontogenic cysts and tumours (Oral Oncol. 2017; 72:165-173).
Response 4: We agree the reasons why some tumors become malignant while others remain benign remains poorly understood. The situation with odontogenic cysts / tumors is an excellent example where tumors have activating mutations but remain benign. We have addressed this example and the publication shortly in the discussion, it now reads:
“Activating mutations (i.e. BRAF, NRAS, etc.) are found in both benign and malignant tumors, i.e. nevi and melanoma. A common theory is the acquisition of additional genetic events lead to tumors progressing, eventually tipping the balance from benign to malignant proliferations. Other events, such as DNA replication errors, have been discussed in odontogenic cysts and tumors which are also mainly benign despite harbouring activating MAP Kinase or CTNNB1 mutations.”
Point 5: Please, clarify and explain the term “advanced” melanoma in the context of the role of CTNNB1 mutations in advanced melanoma. Does that imply a difference to non-advanced melanoma?
Response 5: We thank Reviewer 3 for this critical remark. We agree that we did not adress the term ‘advanced melanoma’ throughout our manuscript and added following paragraph into the methods section on page 3:
“Only melanoma patients with metastatic disease (advanced melanoma) and therefore clinically confirmed malignant disease in stages IIIA or higher were included (see also Table 1).”
Point 6: The authors should avoid inappropriate self-citations (i.e. 1,2,12,14,15,19,20,22,23,24,25,26).
Did you detect inappropriate self-citations by authors
The statistical analysis appears to be sound, the chosen test types are appropriate to the data set. The graphical representations of the results are a useful aid to understanding the text and clearly demonstrate the described trends. The references are from a wide variety of sources and I found no instances of self-citation.
Response 6: We would like to thank Reviewer 3 for addressing the important topic of self-citation. However, we are not completely sure how to interpret this point, as two sentences in the comment seem to contradict each other. From our point of view, self-citation is not a relevant concern within this article. We do refer to and cite the most comprehensive international melanoma studies, especially with regard to genomic examinations and therapy data. Dirk Schadendorf is highly regarded in the field of translational and clinical melanoma research and is involved in many (most) of these international multi-center studies. However, we (Anne Zaremba, Klaus Griewank) are not involved in these publications and thus do not cite ourselves. In our opinion omitting these publications which have defined the melanoma field for the last 10-20 years just because they share one co-author (Prof. Schadendorf) would be erroneous.
In summary, this short article could be of value and is, on the whole, well written. However, several small but significant improvements could, and should, be made.
Response 6: We appreciate this comment and have attempted to adapt the manuscript as best possible to address the points brought up by the reviewer.
References
- Zaremba, A., et al., Genetic and methylation profiles distinguish benign, malignant and spitzoid melanocytic tumors. Int J Cancer, 2022.
- Oulès, B., et al., Clinicopathologic and molecular characterization of melanomas mutated for CTNNB1 and MAPK. Virchows Archiv, 2021.
- Palmieri, G., et al., Molecular Pathways in Melanomagenesis: What We Learned from Next-Generation Sequencing Approaches. Current Oncology Reports, 2018. 20(11): p. 86.
- Liu, D., et al., Integrative molecular and clinical modeling of clinical outcomes to PD1 blockade in patients with metastatic melanoma. Nature Medicine, 2019.
- Gao, J., et al., Integrative Analysis of Complex Cancer Genomics and Clinical Profiles Using the cBioPortal. 2013. 6(269): p. pl1-pl1.

Round 2
Reviewer 1 Report
Overall I am satisfied with the responses of the authors, however still the font within some figures is unreadable and the confidence intervals were not removed from the survival plot as the authors stated in their response.